# Method of Evaluating Multiple Scenarios in a Single Simulation Run for Automated Vehicle Assessment

**DOI:** 10.3390/s23198271

**Published:** 2023-10-06

**Authors:** Inyoung Kim, Donghyo Kang, Harim Jeong, Soomok Lee, Ilsoo Yun

**Affiliations:** 1Department of D.N.A. Convergence, Ajou University, Suwon 16499, Republic of Korea; dlsdud8708@ajou.ac.kr (I.K.); intenr10@ajou.ac.kr (D.K.); 2Department of Transportation System Engineering, Ajou University, Suwon 16499, Republic of Korea; gkfla0731@ajou.ac.kr (H.J.); ilsooyun@ajou.ac.kr (I.Y.); 3Department of AI Mobility Engineering, Ajou University, Suwon 16499, Republic of Korea

**Keywords:** automated vehicle, multi-scenarios, simulation, driving safety, scenario acceleration

## Abstract

With advances in the technology applied to automated driving systems (ADSs), active efforts have been made to evaluate the safety of ADS in various complex situations using simulations. In accordance with these efforts, numerous institutions have developed single-scenario pools that reflect a variety of road and traffic characteristics and ADS performances. However, a single scenario has limitations in comprehensively evaluating the performance of complex ADS. Therefore, this study proposed a methodology that combines and transforms single scenarios into multiple scenarios. This aided in continuously evaluating the ADS performance over entire road segments and implemented this methodology in the simulations.

## 1. Introduction

Because the technology applied to automated driving systems (ADSs) has continuously advanced, many studies have been actively performed to evaluate the safety and performance of ADS. To this end, an evaluation of the safety and performance of automated vehicles (AVs) using scenarios in proving grounds (PGs) or simulations was proposed [1,2]. The scenarios used for AV evaluation are designed to reproduce various situations in PGs or simulations that AVs may encounter during real-world driving. Recently, simulation-based AV safety evaluation is being conducted to reduce the time and cost of evaluation and increase complexity [3,4].

Using scenarios to evaluate AVs makes evaluation more systematic and enables evaluation in a variety of situations. Simulations can generate a wide range of situations that cannot be easily reproduced in PGs or the real world and thus can perform an effective evaluation. Additionally, simulations can repeat the same situation, enabling comparison and the analysis of the performance of multiple systems in the same scenario, and can perform thousands of tests in a short time, making them powerful tools for improving the safety of ADSs [5]. Therefore, several studies mention the importance of developing scenarios for various road sections and situations [6,7,8] and also emphasize how to evaluate driving safety by implementing the developed scenarios in simulations [4].

However, most developed scenarios represent only a single event within a short road section. This implies that it is impossible to evaluate continuous events in long road sections with varying geometric structures. These single-scenario evaluations are useful for safety tests of AVs in specific road sections and various situations; however, resetting scenarios for each road section requires considerable time and effort. For instance, highways consist of main, ramp, and tollgate sections. The use of a single scenario has the limitation of evaluating each section separately. Therefore, to accelerate safety evaluations, it is necessary to transform single scenarios into multiple scenarios that enable continuous evaluation of the entire road section by connecting and combining single scenarios. Multiple scenarios can increase the number of safety evaluations in a short amount of time and intensify AV safety evaluations. It can also include tests with unexpected events due to scenario domain conversion.

In this study, we accelerated the evaluation of AV safety using simulations by transforming single scenarios on highways into multiple scenarios by connecting and combining them. Subsequently, we implemented the proposed multiple scenarios in the MORAI simulation, which is an automated driving (AD) simulation, to confirm whether they are useful for AV evaluations in practice. The remainder of this paper is organized as follows. Section 2 discusses theories and related research on scenarios and simulation standards. Section 3 presents the proposed key definition of multiple scenario generation and consideration for implementing scenarios in the simulation. In Section 4, we generate multiple scenarios based on the proposed definition, implement the generated scenarios in a simulation, and analyze the results to verify the feasibility of using multiple scenarios.

It is expected that using the various scenarios proposed in this study to evaluate the driving safety of AVs will enable the rapid development of AVs in a more realistic evaluation environment.

## 2. Literature Review

### 2.1. Related Theory

#### 2.1.1. PEGASUS Project

Scenario-based evaluation is commonly used to assess the safety of AVs by representing situations that AVs may encounter in the real world autonomously. Germany’s PEGASUS project proposed a scenario development process and methodology to systemically evaluate advanced AD functions. The project defines scenarios in three types: functional, logical, and concrete scenarios [6]. Functional scenarios describe the situations to be evaluated in natural language. They include road networks, static and dynamic objects, and driving situations and have the highest level of abstraction. Logical scenarios specify the types and ranges of functional scenario parameters described in natural language. These scenarios encompass all types of elements that are considered during evaluations. Concrete scenarios are the most specific type of scenarios. They refer to actual experimental values taken from the parameter ranges that are set in the logical scenarios [7].

#### 2.1.2. AV Simulation Standard

The need for simulation-based AV safety evaluations to ensure safety in various environments has increased with advances in AD technology. The Association for the Standardization of Automation and Measuring Systems (ASAM) established a standard format, OpenX, for testing scenarios and simulation-based AV functions. OpenX includes OpenDrive, Open curved regular grid (CRG), OpenScenario (OSC), Open operational design domain (ODD), OpenLabel, and Open open simulation interface (OSI) [9]. OpenDrive defines a static driving environment that supports intersections, road structures, traffic signs, and other information to describe road networks [10]. OpenCRG supports road surface modeling, including potholes and manholes [11]. OSC defines dynamic content related to weather, driving behavior, and objects for AD simulation testing and verification [10]. OpenODD defines a format for representing the operation design domain (ODD) of an AD based on the International Organization for Standardization (ISO) 34503 and the British Standards Institution (BSI) standards. OpenLabel defines labeling methods, labeling structures, and storage file formats for objects and scenarios [12]. Finally, OpenOSI proposes a standard for the interface between AD functions and simulators [9]. Most simulation models for AV evaluation can simulate scenarios written in OpenX format.

### 2.2. Prior Research

In this study, various previous studies evaluating AD functions based on scenarios were scrutinized to extract the conditions to be considered when connecting single scenarios and integrating and adapting multiple scenario simulations.

First, studies that comprehensively evaluated AV functions based on single or multiple scenarios were investigated. Ref. [13] developed a specific framework for developing road segment-based scenarios. The highway main section scenario was developed using the scenario generation procedure proposed in the Pegasus project. This study presented a method for writing functional, logical, and concrete scenarios using traffic accident data. Ref. [14] proposed a scenario development methodology based on the text-mining technique known as term frequency–inverse document frequency. They used descriptive text data on traffic accidents that occurred near crosswalks and intersections on urban roads. They extracted keywords from the text and generated a functional scenario using the extracted word combinations. Ref. [15] analyzed the risk level of surrounding traffic participants using multiple scenarios to evaluate the collision avoidance algorithm for ADS. Risk levels were evaluated under different condition settings. These settings included the driving paths of multiple vehicles, the ego, and targets in both straight roads and intersection scenarios, with varying weather and road surface conditions, such as wet and dry conditions. The indicators selected to evaluate the collision avoidance algorithm were the time to react (TTR), time to collision (TTC), and time to escape (TTE). Ref. [16] implemented various simulation scenarios to accelerate the development of advanced driver assistance systems (ADAS) and ADS. They assessed the reliability of the technology by broadening its range of evaluation. This was achieved by varying the parameter values for single scenarios. The simulation used a virtual test drive (VTD) and evaluated the adaptive cruise control (ACC) or smart cruise control and lane-keeping assist (LKA) functions by constructing straight and curved road sections in the simulation and changing the speed and position of the ego and target vehicles. The study mentioned that it is possible to evaluate ADAS functions quickly and efficiently and that there is a significant effect in reducing evaluation costs. Ref. [17] proposed a cooperative caching scheme in the VEC (vehicular edge computing) based on asynchronous federated and deep reinforcement learning (CAFR). The proposed scheme was evaluated and verified via an urban road simulation environment. Via the evaluation, it was determined that the CAFR scheme was superior to other baseline caching schemes.

Second, we conducted a comprehensive review of the existing literature, employed the data collected to generate AD evaluation scenarios, and subsequently classified and prioritized them based on their perceived level of difficulty. Many researchers are interested in extracting important scenarios. Ref. [18] presented a testing scenario library generation (TSLG) framework, which is a comprehensive library of testing scenarios designed to address the limitations of PG testing. They computed the parameter values to be tested in these scenarios by assigning importance scores to all possible values within the ODD. The parameter values for the importance scores were the maneuver challenge and the exposure frequency of the scenario. Fifty-seven important scenarios, accounting for 1.67% of the total, were extracted. Similarly, [5] proposed a test-scenario generation framework for evaluating the driving convenience of AVs. They generated a scenario library from naturalistic traffic data and used occurrence frequency and random forest models to derive the scenarios. Critical scenarios were extracted based on the comfort scores. Critical scenarios were found to be more challenging than existing scenarios. In addition, a clustering algorithm was used to prevent scenario overlap. Ref. [19] measured the similarity between scenarios generated by vehicle data and those classified using a random forest. The scenario similarity was derived by comparing the number and positions of the target and ego vehicles in the scenario. Scenario classification was used to broaden the test scope and diversity and provide a method to avoid performing the same test. Various studies have also attempted to derive critical scenarios by combining scenarios based on ODD. Ref. [20] emphasized the need for diverse and specific experimental value combinations via ODD when designing test scenarios. Scenarios were generated by combining ODD parameters with a complexity index. Subsequently, the authors created a simulation-based framework to examine the environmental conditions that can lead to AV errors and thoroughly evaluated the performance of AV functions.

The main objective of this study was to continuously evaluate AV functions by implementing multiple scenarios in a single simulation. Therefore, scenario-based evaluations of the AV functions using simulated environments were conducted. Ref. [21] proposed a virtual-environment-based AV function evaluation framework that can objectively and quickly evaluate advanced AD functions. They formulated complicated scenarios and developed automated algorithms that generated evaluation metrics for testing the AV functions. There were four different scenarios, which included pedestrians in a straight section under rainy conditions, cyclists and target vehicles in an intersection, and target vehicles in a curved section. A total of 12 indices were used to evaluate the four aspects of the AV function. They included driving safety, ride comfort, intelligence, and efficiency. The study demonstrated a high consistency between the evaluation in a virtual environment and that in a real environment.

### 2.3. Lessons Learned

Numerous studies were conducted to systematically develop single scenarios that verify vehicle behavior on short road sections. However, only a few studies have focused on developing multiple scenarios that can systematically evaluate multiple events on long road sections with various geometries. Although some researchers attempted to integrate multiple scenarios for evaluation, they evaluated a single situation by merely changing the parameter values to test specific ADAS functionality. In addition, although some multiple scenarios were implemented to evaluate ADAS functions, a full-scale multiple-scenario development procedure and multiple-scenario verification were not performed.

The scenario integration methods used in these studies simply connected straight and curved sections or set different environmental conditions, such as rain and clear weather. It is believed that the importance of single scenarios and multiple scenarios was not perceived to be significantly different in the past, and no comparison of the differences between the two types of scenarios was made. Recent studies have focused on assessing safety based on scenarios similar to real road conditions to evaluate complex AV functions. Therefore, it is believed that research focusing on multiple scenarios will become increasingly important.

The aforementioned studies solely considered road types or weather conditions without proposing any systematic approach for combining single scenarios into multiple scenarios. Consequently, creating multiple scenarios that accurately reflect the actual world is inadequate. In this study, we proposed a specific framework for developing multiple scenarios capable of replicating real-world conditions, including roads, traffic, weather, and environmental factors. The scenarios were formulated using the functional, logical, and concrete scenario systems defined within the PEGASUS project. We ensured that each stage of the scenario was supported by a meticulous and comprehensive set of procedures to ensure the highest level of detail and accuracy. In addition, this study proposed a novel process for implementing and validating these multiple scenarios within simulations, which was not previously discussed in the literature. We present a systematic and specific procedure for the development of multiple scenarios and the verification of their accuracy via simulations. This study validated the reliability of these scenarios, thereby enabling an efficient safety evaluation of AD based on scenario-based testing.

## 3. Methodology

### 3.1. Overview

The main objective of this study was to produce a set of multiple scenarios by generating single scenarios and connecting them sequentially. This approach aims to address the inherent limitations of single scenarios when verifying vehicle behavior on specific road sections. Additionally, the generated multiple scenarios are implemented in ADS to demonstrate the effectiveness of multiple scenarios by comparing the performance of single and multiple scenarios.

To this end, a method of converting from a single scenario to multiple scenarios and a method of implementing and evaluating multiple scenarios within a simulation were presented according to the scenario development procedure (functional, logical, concrete) presented in the PEGASUS project. Details on this are as follows:Selecting multiple functional scenarios: select single functional scenarios suitable for simulation and create a multiple functional scenario by connecting single functional scenarios.Generating multiple logical scenarios: create a logical scenario by expanding elements and scope on the single functional scenarios selected in the previous step.Generating multiple concrete scenarios: determine specific experimental values from the elements and scope of the logical scenario.Validating the multiple concrete scenarios: input and evaluate multiple concrete scenarios in simulation.

In this section, specific methods and conditions for converting and creating multiple scenarios and implementing the scenarios in simulation according to the above procedure are presented. The simulation employed the Mobility Research AI(MORAI) simulation, which used the ASAM OSC 1.2 version. The overall procedure for generating and verifying multiple scenarios in this study is shown in Figure 1.

### 3.2. Conversion of Single Scenarios into Multiple Scenarios

#### 3.2.1. Multiple Functional Scenario

The conversion from a single scenario to multiple scenarios occurs at this stage. Prior to conversion to a multiple functional scenario, the difficulty of the scenario must first be estimated to extract a single functional scenario suitable for simulation. The reason for estimating the difficulty level of a single functional scenario is to expand the case diversity when creating multiple scenarios. In the new assessment/test methods (NATMs) from the validation method for automated driving (VMAD) of Working Party 29 (WP.29) [22], AV evaluation methods are classified as simulation, PG, and real-world testing. Furthermore, scenario characteristics are classified as edge case, critical, and typical for each evaluation method. Edge cases are scenarios with high risks and low probabilities of occurrence and are suitable for simulation. Critical scenarios have lower risks and higher probabilities of occurrence than edge cases and are suitable for PG testing. Typical scenarios have the lowest risk and highest probability of occurrence, making them suitable for real-world testing [22].

In this study, we estimated the difficulty level of the single functional scenarios required for simulation testing by using the frequency and severity of traffic accidents [13]. The frequency of traffic accidents was determined by the number of accidents related to the scenario among the traffic accident information in the functional scenario data. The severity of traffic accidents was obtained by applying equivalent property damage only (EPDO) to the type of traffic accident in the reference scenario. The EPDO converts the damage resulting from an accident into a unit that represents its severity based on the weights assigned to different types of accidents (fatal, injury, and property damage) [23]. In Korea, the weights by accident type are 12 for fatal accidents, 3 for injuries, and 1 for property damage [24,25]. Only accidents that resulted in personal damage such as fatalities and injuries were considered in this study. The EPDO representing accident severity was computed as follows in Equation (1). Where Fatal is the number of fatal crashes and Injury is the number of injury crashes. The difficulty level of a scenario was estimated using K-means clustering with accident frequency and severity as input parameters. K-means clustering is an unsupervised learning algorithm that assigns each data point to a group based on a given number of clusters, K, and initial values. This technique performs grouping by analyzing similarities by measuring the distances between given observations without relying on prior information regarding clear categories [26,27].
(1)EPDO = (Fatal×12)+(Injury×3)

Next, connection conditions were defined to convert the extracted single functional scenario into a multiple functional scenario. The actual driving environment and situations were reflected in multiple scenarios. If scenarios are connected to a lack of realistic traffic flow and vehicle dynamics, it may be difficult to implement plausible scenarios in the simulation, and proper evaluation cannot be conducted owing to the accompanying errors. In this study, the prerequisite conditions for connecting single scenarios were defined to facilitate the immediate implementation of multiple scenarios. By connecting the single scenarios schematically depicted in Figure 2, we listed the key prerequisites for considering the road environment and vehicle conditions.

The four main conditions that were considered in connecting the scenarios are as follows:Reflecting the characteristics of the number of lanes—If the number of lanes, which is a road design condition, is not reflected (e.g., when connecting a 4-lane highway main section scenario with a 2-lane toll gate section scenario), congestion can occur even in free-flowing traffic because of the decrease in the number of lanes. This does not reflect the traffic flow between the main highway section and the toll gate section in free-flowing traffic. In addition, there is a risk of collision accidents owing to frequent lane changes caused by surrounding vehicles.Matching the ego vehicle and scenario direction—If the type of ego vehicle is different or the direction of the scenario is not seamless, consecutive scenario experiments cannot be conducted, resulting in longer experimental times or failures in scenario connection.Generating multiple events—When scenarios repeat the same events, safety verification can be less reliable because the ability to respond to multiple situations cannot be evaluated.Traffic flow similarity—If the traffic flow deviations between scenarios are large, the interactions of the ego vehicles with the speed and behavior of the target or surrounding vehicles can change suddenly, leading to unnatural driving behavior.

Therefore, connecting scenarios without considering these conditions limited their implementation in the simulation. In this study, the aforementioned conditions were classified as static and dynamic elements for ease of presentation. Static elements refer to elements that do not change during the simulation runtime, such as hyperparameters or configuration files for the initial conditions, whereas dynamic elements refer to elements that can change during the simulation runtime. The considerations when connecting scenarios according to the environmental and vehicle conditions are presented in Table 1.

#### 3.2.2. Multiple Logical Scenario

The converted multiple functional scenarios should be specified in a form applicable to the simulation. To express the situation clearly, the selected functional scenario was transformed into a logical scenario based on these considerations. Logical scenarios were created by defining the scenario elements and scope. This study defined a scenario scope that was compatible with the simulation by utilizing the ODD classification and elements presented in the OpenODD of the ASAM. ODD comprises various parameters that can affect AD operations, such as road type, weather conditions, and traffic situations, as safe operating conditions for AVs [28]. Recently, the ISO established ODD standards to unify the expression of ODD, and OpenODD defined ODD based on a standard, BSI, ISO, and automated vehicle safety consortium (AVSC) [29]. OpenODD classified ODD details into three categories: scenery, environmental conditions, and dynamic elements. The scenery includes 2D and 3D road spaces where AVs drive, and the environmental conditions include factors unrelated to roads, such as weather, particles, luminance, and connectivity. Dynamic elements include objects, such as road users and ego vehicles, that directly affect road safety evaluations [30]. The elements in OpenODD can be transformed into a simulation-compatible language that facilitates the implementation of scenarios.

In this study, the elements and scope to be included in the logical scenario were proposed based on the scenery, environmental conditions, and dynamic elements corresponding to OpenODD classification. The elements and scope were derived from the ODD of MORAI simulation, which is based on the ASAM OSC 1.2 version. The ASAM OSC 1.2 version includes dynamic objects, weather, and speed conditions, while MORAI simulation includes various elements and scopes such as road geometry, communication, time, and signage in addition to those in ASAM OSC.

Subsequently, the logical scenario should be able to generate numerous scenarios via the combination of elements and scope. Therefore, the elements and scope of the logical scenario were classified into variable elements that can be changed in the simulator and fixed elements. The combination of variable elements enables the generation of numerous scenarios. The procedure for generating logical scenarios is illustrated in Figure 3, and the proposed scenario elements and scope are elaborated on in Section 4.3.2.

#### 3.2.3. Multiple Concrete Scenario

Concrete scenarios are generated by combining the variable values within logical scenario elements, resulting in hundreds of thousands to tens of millions of scenarios. These concrete scenarios can provide actual experimental values that can be converted into an applicable form for simulations. Various situations ranging from simple to complex can be implemented by combining numerous elements. At the moment, the actual experiment value can be exploited with random sampling.

### 3.3. Multiple Scenario Implementation

#### 3.3.1. Establishment of Test Conditions

To implement the scenario in the simulation, the experimental values extracted from the concrete scenario must be converted into xosc format supported by OSC. However, it is important to follow certain considerations to avoid situations in which unnatural driving behavior or scenario implementation owing to collisions cannot be realized during the scenario setting.

We formulated several factors while applying the exploited concrete scenario experimental values to the MORAI simulation after the OSC format conversion. The variables defined in OSC correspond to the variable ODD of the logical scenario parameters. OSC consists of the ‘Init’ and ‘Story’ stages and these stages derive the experimental conditions and considerations. Init is the stage for setting the scenario environmental conditions, such as initiating the state of the objects, including the ego vehicle, and its interaction with other objects, weather, traffic, and signal information. The story is a stage that defines the events that occur during the simulation and the behaviors of the objects when each event is activated. This stage includes a start/stop trigger, maneuver group, actors, and maneuvers (event, action). Triggers establish the criteria for initiating certain scenario components and dictate when actions (which control the entity’s longitudinal and lateral movements) should start or stop.

At the Init stage, there are several factors to consider regarding the speed of dynamic objects. For instance, it is necessary to set the speed of a vehicle differently, depending on the trigger agent and maneuver of the vehicle. AVs reflect the driving behavior of the operators. Therefore, when evaluating lane changes or overtaking scenarios, the ego-AV speed should be higher than the target vehicle speed. Conversely, the ego vehicle speed should be the same as or slower than the target vehicle speed in the car-following scenario. Thus, the speed range should be considered depending on the trigger agent and maneuver of the vehicle.

We have derived key conditions in the story stage to simulate dynamic objects that behave similarly to a driver model. For example, it is necessary to define the acceleration/deceleration and arrival positions of the target and ego vehicles in advance to control their movement when an ego vehicle changes lanes or avoids obstacles. In addition, a sufficient driving distance and time interval should be allocated to cope with transitioning events for consecutive events. This allocation overcomes any discontinuities that may occur during scenario connections.

#### 3.3.2. Selection of Evaluation Metrics

To compare and evaluate the performance of a single scenario and multiple scenarios, time to collision (TTC) was selected as an evaluation metric. Various metrics such as time to escape (TTE), time to react or time to region (TTR), and time to steer (TTS) can be used for performance verification in an AD assessment scenario [15,31]. However, TTC is the commonly used metric when evaluating scenarios and is mainly used to evaluate vehicles following scenarios [13,32]. TTC is easy to apply to complex scenario analysis [33]. For lane changing, TTS is mainly used. In this study, TTC was used to evaluate the vehicle following the scenario in the tollgate section.

TTC refers to the time required for two vehicles to collide if they continue driving at their current velocity. A lower TTC value indicates a higher severity of the collision, and a zero TTC indicates that a collision has occurred. The TTC generally has a threshold value, where values above the threshold indicate lower severity of collision and vice versa. Therefore, a higher TTC value suggests that the ADS operates more stable [34]. The TTC was computed using Equation (2) as follows:(2)$$TTC\;=\;\frac{X_{L}-X_{F}}{V_{F}-V_{L}},$$
where $X_{L}$ and $X_{F}$ represent the positions of the lead and following vehicles, respectively, and $V_{L}$ and $V_{F}$ represent the speeds of the lead and following vehicles, respectively [34].

## 4. Generation and Validation of Multiple Scenarios

### 4.1. Overview

In this section, we describe the process of generating multiple scenarios and verifying them using the MORAI simulation based on the methodology proposed in Section 3. Using the methodology proposed in the previous section for generating and implementing multiple scenarios, we performed the entire procedure, from converting single scenarios to multiple scenarios to simulation-based verification. As a simulation-based scenario verification method, we converted the concrete scenario created in the last stage of scenario generation into an OSC format and implemented it in the simulation. We then compared the results of the single scenario simulations with those of the multiple scenario simulations. Via this process, we demonstrated the performance of multiple scenarios.

### 4.2. Site Selection and Scenario Collection

The purpose of this study was to develop multiple scenarios for highways. Thus, we designated the spatial range of the K-City highway map, an AD testbed with a realistic evaluation environment. First, we collected single scenarios developed for each segment, such as the ramp, main, and toll gate sections. A pool of functional scenarios for highways was collected based on traffic accident data developed by [13]. The collected scenarios were written in natural language and consisted of “scenario description” (position of vehicles, driving path of the ego, target, surrounding vehicles), “road geometry” (number of lanes, lane type), “movement objects” (ego vehicles, surrounding vehicles), and “accident information” (accident frequency and severity). The scenario pool contained 59 scenarios, comprising 22 ramp sections, 29 main sections, and eight toll gate sections. The scenarios were classified into nine categories: car-following, lane deviation, stopping, cut-in, cut-out, cut-through, wrong-way driving, driving, and lane changing.

### 4.3. Multiple Scenario Generation from Single Scenario

#### 4.3.1. Functional Scenario Generation

In this section, we classified scenarios using K-means clustering to derive suitable single scenarios for testing in the simulation environments. Our K-means clustering model used data on the frequency and severity of traffic accidents. To calculate the severity of traffic accidents, we applied the EPDO formula to the fatalities and injuries data. The number of clusters (K) was set to three, considering the scenario types (edge case, critical, and typical) mentioned in Section 3.2.1. Based on the results of the K-means clustering, the difficulty level of a single scenario can be categorized as follows:an edge case scenario if traffic accidents are rare, but the severity is high;a critical scenario if traffic accidents occur frequently, but the severity is not high;a typical scenario if traffic accidents are rare and the severity is low.

Figure 4 shows the classification results for the three scenarios in a highway section, where the X-axis represents accident frequency, the Y-axis represents traffic accident EPDO, and the star notation represents the centroids for scenario classification.

The circle around the star represents a scenario, where the yellow circle represents an edge case, the purple circle represents a typical case, and the green circle represents a critical scenario.

The Silhouette index was used to evaluate the fitness of the K-means clustering model. Silhouette is a clustering validity index that measures the similarity of each data point in one cluster to other data points in the same cluster, compared to data points in other clusters [35]. Generally, a Silhouette index greater than 0.5 is considered a well-clustered result [36]. In the clustering results, the Silhouette indices of the ramp, main, and tollgate sections were 0.54, 0.60, and 0.51, respectively. Because all the indices were greater than 0.5, the model was validated.

Based on the estimated scenario difficulty level, ten edge-case scenarios were identified, including four scenarios in the ramp section, five scenarios in the main section, and one scenario in the toll gate section. All ten scenarios were deemed suitable for testing in the simulation. After extracting scenarios for each highway section corresponding to the edge cases, the connectivity was verified based on the connection conditions specified in Table 1. The highway section scenarios that satisfied all the connection conditions are listed in Table 2. The specific situations for the selected multiple functional scenarios are as follows: evaluation of perception and response to front-end congested vehicles in the ramp section, evaluation of driving safety for ego vehicle cut-in in the main section, and evaluation of car-following after target vehicle cut-in at the toll gate section.

#### 4.3.2. Logical Scenario Generation

The logical scenario makes the functional scenario more detailed so that the experiments can be conducted in diverse ways. In this study, we presented detailed scenario elements based on the OpenODD classification and defined scenario elements that are feasible for the MORAI simulation. In addition, we divide the ODD into variable and fixed elements to increase the usability of the logical scenario. The selected functional scenarios were changed to logical scenarios based on the ODD classification and elements. As a result, thirty-eight elements were selected for the entire highway section scenario, and 13 variable elements were derived and utilized when generating concrete scenarios. Via this process, experimental values that could be implemented in the simulations were obtained.In this section, the logical scenario elements were set by the defined ranges in the MORAI simulation ODD and the OSC parameters. Different increasing ranges (i.e., ΔValue) were set for each element. The range for the ‘illumination’ factor was set to an increasing range of 2, which was set by the MORAI simulation. ‘Varied’ in variability refers to the changeable element in the MORAI simulation. Table 3 lists the logical scenario elements and ranges defined for the highway ramp section.

#### 4.3.3. Concrete Scenario Generation

We scrutinized 13 variable elements from logical scenarios that could be implemented on highway sections. Subsequently, we generated all possible combinations of scenario elements to create scenarios. However, traffic volume and speed varied according to the density; therefore, they were excluded from the variable combinations. Finally, we found variable elements 11 for the ramp section, 12 for the main section, and 13 for the toll gate section. For the ramp section, combining 68 variable values resulted in 2,509,056 scenarios. For the main and toll gate sections, combining 78 experimental values resulted in 12,773,376 concrete scenarios.

Concrete scenarios were extracted from the ramp, main, and toll-gate sections of the highway using random sampling techniques. The extracted scenarios were categorized into two types: free-flow and congested traffic conditions. These types were implemented and verified via simulations. However, the simulated weather, particles, and traffic flow were designed to closely match the actual conditions to reflect real-world scenarios. The extracted final concrete scenario for the logical scenario categorized as “varied” is shown in Table 4.

### 4.4. Multiple Scenario Implementations and Validations

#### 4.4.1. Scenario Implementations Based on Setting Test Conditions

To implement the concrete scenarios in the simulation, the experimental setup conditions discussed in Section 3.3.1 should be satisfied. Therefore, it was necessary to adjust the speed of the dynamic objects, driving position, and delay time to provide a buffer for the next event of the scenario elements. The final setting parameters were converted into OSC format.

To reflect the characteristics of the short road section of the K-City highway, the traffic volume was designed to be 40% lower than the actual road design speed at 42, 72, and 72 km/h for free-flow situations, and the ego vehicle speed was set higher than the target vehicle speed when entering the main section from the ramp section or changing lanes on the main section. The driving paths of the ego and target vehicles were set in advance using an empirically located lane, assuming that the vehicles followed the lane properly. To ensure that the ego vehicle could respond continuously to the next event, a five-second buffer was placed between each event to secure the driving distance. In congested traffic conditions, the speed was set to 40% lower than the actual road design speed at 36 km/h, 57 km/h, and 57 km/h, whereas speed changes, driving paths, and transition time buffers were set to be the same as in free-flow situations. In addition, various other hyperparameters, such as the initial speed and acceleration/deceleration of the vehicle, distance, execution time, driving trajectory, and weather information, were set to implement the scenario.

The MORAI simulation is an AD simulation platform that provides a virtual driving test environment similar to the actual environment using x-in-the-loop (XIL). The simulation provides a digital twin similar to the driving environment based on the vehicle, weather, lighting, and obstacles. It also offers scenarios that are difficult to reproduce but are occasionally encountered on real roads. MGeo is MORAI’s proprietary format for precision mapping. The conversion of simulation scenarios into the OSC format was achieved using the MORAI scenario runner. The MORAI scenario runner loads the scenario data in the form of xosc files that are defined based on the OSC format and performs the concrete scenario by linking them to the MORAI simulation. The MORAI scenario runner can control vehicles and objects inside the MORAI simulation, and the scenario is defined based on a map in MGeo format.

The experimental values of concrete scenarios that satisfy the predefined test setting conditions, as shown in Figure 5, were converted into xosc files. These files can be applied in the simulation to implement the scenarios using the MORAI scenario runner.

We finally implemented the scenarios presented in Table 4 in the simulation by applying two different environmental conditions. Specifically, in the highway ramp section, the ego vehicle recognized and performed a lane change to the main section where the stopped target vehicle was located. Subsequently, the ego vehicle recognized the surrounding vehicles in the main section and performed another lane change to enter the toll gate. Finally, the ego vehicle recognized and controlled a sudden cut-in target vehicle at the toll gate entrance. Two scenarios were implemented: scenario 1 represented free-flow traffic conditions (level of service (LOS) A) in the evening and rainy weather, and scenario 2 represented congested traffic conditions (LOS C) on a clear day during daytime. Under free-flow traffic conditions (LOS A), three surrounding objects were added around the ego vehicle, whereas under congested traffic conditions (LOS C), approximately 11 types of objects were added around the ego vehicle. All vehicles in the scenarios were designed to follow a mixed traffic flow of ACC and non-ACC vehicles. Figure 6 shows the cases where multiple scenarios were implemented in the simulation.

#### 4.4.2. Compare Single and Multiple Scenarios

To verify the performance for multiple scenarios, multiple and single scenarios were compared via simulations. Both scenarios involved similar situations.

In this study, the TTC was obtained in the toll gate section, which included a scenario in which the ego vehicle followed the target vehicle to evaluate the performance in multiple scenarios. The threshold value for the TTC was set to 1.5 s, as suggested by [37,38]. The ramp section was excluded from the evaluation because the single and multiple scenario conditions were identically configured. Additionally, since the main section is mainly a scenario where the ego vehicle changes lanes, it was excluded due to limitations in evaluation using TTC. The speed and position of the ego vehicle and the speed and position of the target vehicle directly interacting with the ego vehicle were extracted to compute the TTC. The analysis of the TTC for hazardous situations below the threshold value showed that in free-flow situations, the TTCs for multiple and single scenarios were 1.04 and 1.18 s, respectively, with the TTC for the multiple scenarios being 0.14 s lower. In congested situations, the TTCs for multiple and single scenarios were 0.69 and 0.83, respectively, with the TTC for multiple scenarios being lower.

Based on these results, it can be concluded that the randomness of the maneuvering behavior of the surrounding or target vehicles increased as the multiple scenarios transitioned from consecutive scenarios. During congestion, there was a queue owing to a decrease in the number of lanes from the main section of the highway to the toll gate, resulting in a sudden interruption. Based on this case, it was demonstrated that the ego vehicle suddenly applied the brakes to avoid collision. Figure 7 compares the TTC results between the multiple and single scenarios in the toll gate section. X-axis presents time, and y-axis means TTC in seconds.

Therefore, the strength of the AD safety assessment is higher in multiple scenarios than in single scenarios, as it creates more dangerous situations and generates various unexpected events. In particular, according to [39], the ability to respond to potential collisions of AV has become more important; therefore, the need for multiple scenarios proposed in this study is expected to be emphasized in the future. In addition, multiple scenarios have more benefits beyond increasing the robustness of safety evaluations. When converted into multiple scenarios, safety evaluations can be performed continuously across the entire road segment rather than specific sections, reducing the time required for evaluation. Furthermore, the ability to evaluate scenario connections and various parameter combinations contributes to accelerating safety evaluations. Thus, the multiple scenarios proposed in this study are considered more advanced than single scenarios and can be considered the optimal solution for the safety assessment of complex AV functions.

## 5. Conclusions

In this study, we proposed a scheme for integrating multiple scenarios that can be continuously evaluated throughout an entire road section by connecting and combining single scenarios based on specific conditions. Our goal was to accelerate safety evaluations, increase their intensity, and make AV system development more reliable and secure. By adapting the scenario development process (functional, logical, and concrete scenarios) of the PEGASUS project, we developed specific methods and procedures for implementing scenarios in simulations to enable rapid AV safety evaluation.

We derived the connection conditions and scenario representation methods, as well as the experimental setup conditions within the simulation. These convert from a single scenario to multiple scenarios. Based on this, multiple scenarios (functional, logical, concrete) are created and then verified scenario performance via simulation.

Based on the aforementioned conditions, two multiple scenarios were created and implemented in the simulation (a free-flow traffic situation during the evening and a congested situation during the day). Both single and multiple scenarios were implemented in the simulation, which used similar situations, and the two scenarios were evaluated based on the TTC. The evaluation of multiple scenarios showed superior results compared to single scenarios, as indicated by an enhanced time-to-collision (TTC) of 0.14 s. In addition, the multiple scenarios generated unexpected events as they continued from the current scenario to the next. The generated unexpected events with multiple scenarios increase the safety and stability evaluation of AV by creating riskier or more unexpected situations than single scenarios.

The limitations of this study and future research directions are as follows. The scenarios were clustered based on the accident frequency and severity. However, there is a limit in the application of the proposed methodology when the accident frequency or severity is not provided in the developed scenarios. In future work, it will be necessary to scrutinize a methodology to approximate these requirements if they are not specified. Additionally, compared to a single scenario, multiple scenarios are demonstrated to reduce overall experiment time and effort due to the reduction in the number of scenario-based evaluations. However, future research on how much time is reduced will be proven via additional experiments bringing more scenarios.

## Figures and Tables

**Figure 1 sensors-23-08271-f001:**
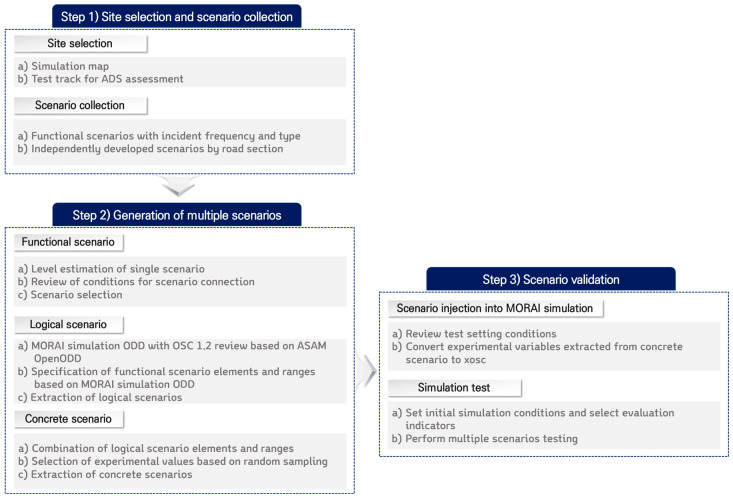
Framework for generation and validation of multiple scenarios.

**Figure 2 sensors-23-08271-f002:**
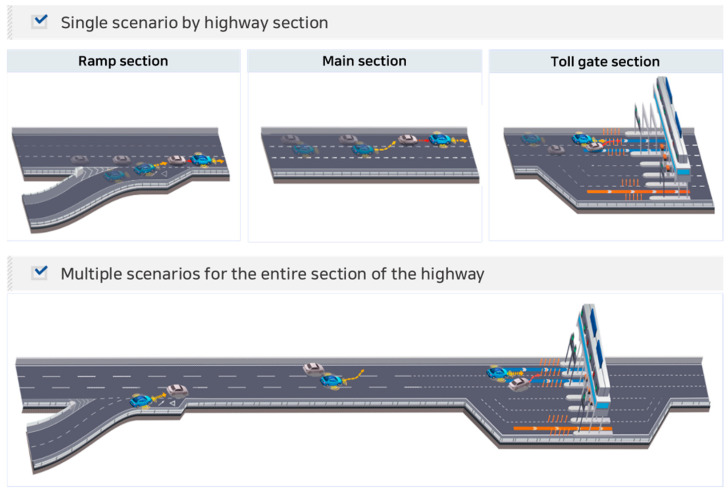
Example diagram of expressway scenario.

**Figure 3 sensors-23-08271-f003:**
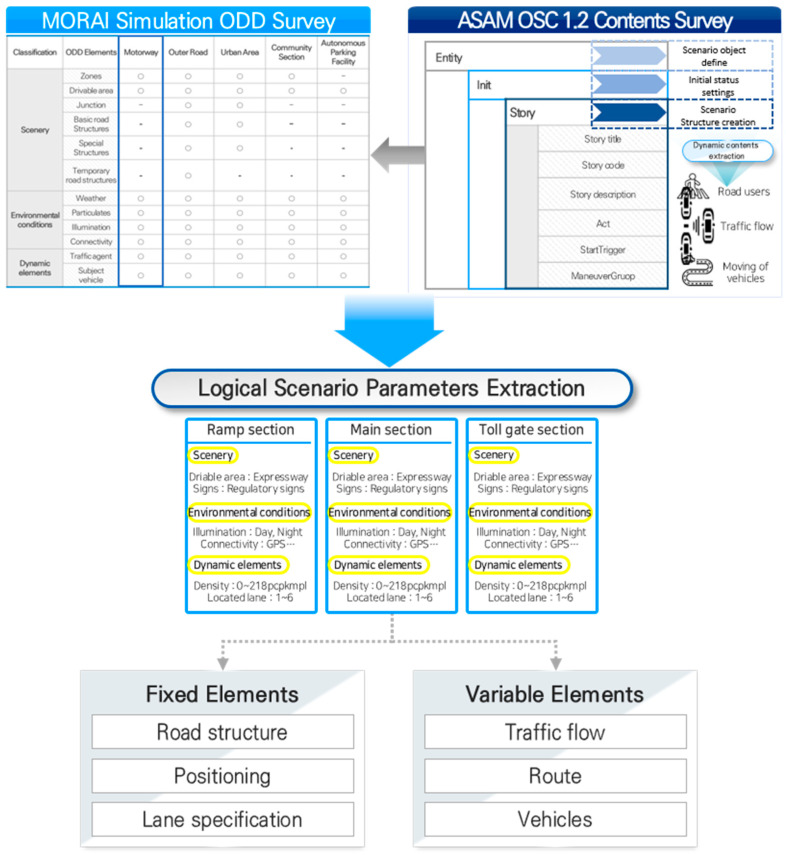
Procedure for deriving logical scenario elements.

**Figure 4 sensors-23-08271-f004:**
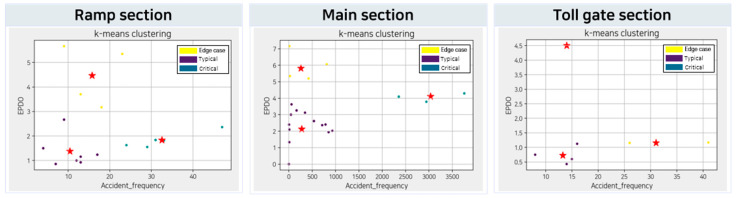
K-means clustering results for single functional scenario classification.

**Figure 5 sensors-23-08271-f005:**
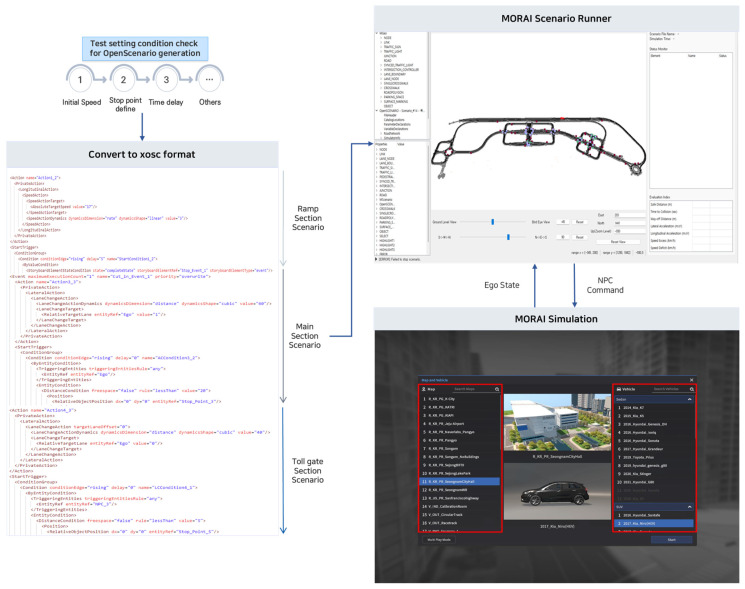
Concrete scenario implementation procedure considering test setting conditions.

**Figure 6 sensors-23-08271-f006:**
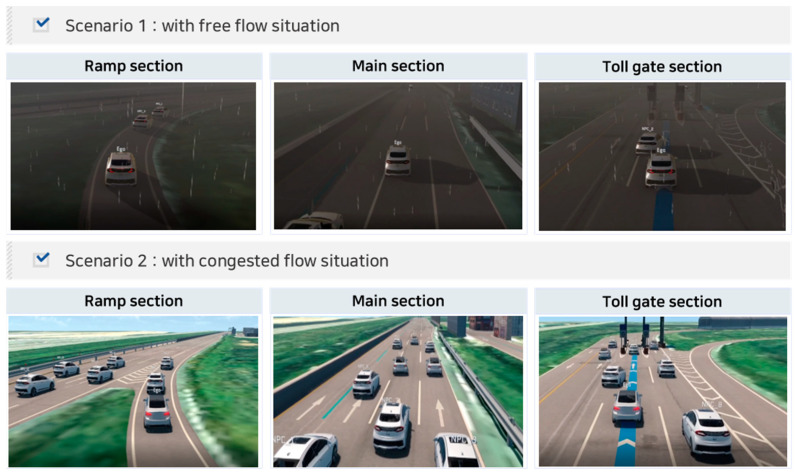
Implemented multiple scenarios adapting test condition parameters.

**Figure 7 sensors-23-08271-f007:**
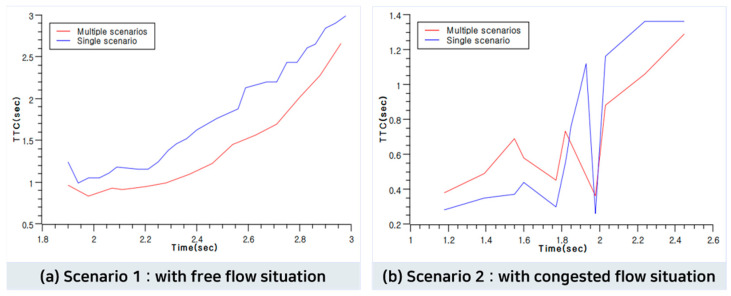
TTC results of multiple and single scenarios.

**Table 1 sensors-23-08271-t001:** Considerable features for scenario connection.

Classification	Static Element	Dynamic Element
Environment condition	The characteristics of the number of lanes for each road section should be reflected.	The direction of scenario should be the same.
Vehicle condition	The ego vehicle to be evaluated must be the same. Traffic flow between scenarios should be similar.	Ego vehicle movements should not be repeated equally. The ego vehicle must have the same driving path.

**Table 2 sensors-23-08271-t002:** Single functional scenarios satisfying connection conditions.

Classification		Ramp Section	Main Section	Toll Gate Section
Environmental condition	Number of lanes	2	3	6
	Direction of scenario progress	E	E	E
Vehicle condition	Ego vehicles	A1	A1	A1
	Target vehicles	2	1	1
	Position	1	2	2
	Driving path of the ego vehicle	W → E	W → E	W → E

E (east): Direction of scenario; A1 (A1, A2, B1, B2 …): Vehicle number; W → E (from west to east): Driving direction of ego vehicle.

**Table 3 sensors-23-08271-t003:** Logical scenario elements and ranges in the ramp section.

Classification	ASAM OpenODD		Logical Scenario with MORAI Simulation ODD			
			Elements	Range		Variability
Scenery	Zones	Regions or states	Hwaseong-si, Gyeonggi-do, Republic of Korea			Fixed
	Drivable area	Type	highway, slip roads			Fixed
		Geometry	Horizontal plane	Straight, curve		Fixed
			Transverse plane	Divided		Fixed
				Pavements (Asphalt)		Fixed
				Superelevation/banking		Fixed
			Longitudinal plane	Up-slope, down-slope, level plane		Fixed
		Lane specification	Lane dimensions	3.5 m		Fixed
			Lane marking	Solid yellow, curb bus lane, lane changing, Hipass (electronic toll collections system), safe zone, edge, median		Fixed
			Lane type			Fixed
			Number of lanes (Two-way)	Ramp	1	Fixed
			Direction of travel	One-way		Fixed
		Signs	Regulatory signs	Not applicable, speed limit, lane control		Varied
		Edge	Line markers			Fixed
			Solid barriers	Concrete guardrail, plastic		Fixed
			Temporary line markers	Attachable		Fixed
		Surface	Dry, wet			Varied
	Fixed road structures		Street furniture (e.g., bollards)	Not applicable, tubular markers, Chevron alignment sign, speed bump, raised pavement marker, crash cushion, noise barrier, median barrier		Fixed
	Special structures		Tunnels			Fixed
			Toll plaza			Fixed
	Temporary road structures		Not applicable, construction section			Varied
Environmental conditions	Weather		Sunny, cloudy, snowfall, rainfall			Varied
	Particulates		Not applicable, water drop, fog			Varied
	Illumination		Day	5–11 (ΔValue: 2)		Varied
			Night	12–23 (ΔValue: 2)		Varied
			Cloudiness	Sunny, cloudy		Varied
	Connectivity	Positioning	GPS			Fixed
		Fleet management	Enterprise management system	CCTV, RSU, signal controller, incident detection, traffic signal, Evaluation system		Fixed
		Obstruction	Not applicable, GPS jamming system			Varied
Dynamic elements	Agent type	Special vehicles	Not applicable, ambulance, police			Varied
		Basic vehicles	Not applicable, motor vehicle, non-motor vehicle, semitrailer, trailer, train, tram, truck, van, vulnerable road users (children male/female), adult male/female, cyclist, policeman, obstacle (CargoBox, RedBarrel, TrafficBarrel, WoodBox, YellowBarrel, NCAP_GVT), animal			Varied
		Attributions	Density of agents	0–22 (pcpkmpl) (ΔValue: 1)		Varied
			Speed	0–70 km/h (ΔValue: 10)		Varied
	Subject vehicle	Maximum speed	0–70 km/h (ΔValue: 10)			Varied
		Route	Moving	Lateral	Straight, cut-in/out/through	Fixed
				Longitudinal	Constant, acceleration, deceleration, rest	Fixed
			Located lane	1		Varied
		Weight	Customization (unit: kg)			Fixed

**Table 4 sensors-23-08271-t004:** Concrete scenario extraction results.

Concrete Scenario #1 with Free Flow Situation (#2 with Congested Flow Situation)				
Parameters		Value		
		Ramp Section	Main Section	Toll Gate Section
Signs	Information signs	-	-	Hipass (Exit)
	Regulatory signs	Not applicable	Speed limit	Not applicable
Surface	-	Wet (dry)	Wet (dry)	Wet (dry)
Temporary road structures	-	Not applicable	Not applicable	Not applicable
Weather	-	Rain (sunny)	Rain (sunny)	Rain (sunny)
Particulates	-	Not applicable	Not applicable	Not applicable
Illumination	Day	20 (11)	20 (11)	20 (11)
Connectivity	Obstruction	Not applicable	Not applicable	Not applicable
Agent type	Special vehicles	Not applicable	Not applicable	Not applicable
	Basic vehicles	Motor vehicle	Motor vehicle	Motor vehicle
	Density of agents	4 (18)	5 (14)	5 (14)
Subject vehicle	Located lane	1	2	2

## Data Availability

The data presented in this study are available on request from the corresponding author.

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
