# Peer review of "Method of Evaluating Multiple Scenarios in a Single Simulation Run for Automated Vehicle Assessment"

_sensors, 2023, doi:10.3390/s23198271_

Round 1

Reviewer 1 Report

The authors were required to concentrate on the below comments.

1.        Some grammatical, spelling mistakes, and annotation errors can be seen in the manuscript. A careful revision is required.

2.        The literature presented in the introduction and literature review section is too lengthy and also has some unnecessary content. It needs to be made clear with more references and quality.

3.        The figures presented in the manuscript were not of good quality, which should be improved.

4.        Figures 1, 3, and 5 are to be in a clear manner and need to be described for better understanding for readers. Unnecessary things to be eliminated from figures and as well as quality should be enhanced.

5.        In Figures 4 and 7, the graphs provided are to be described in every aspect and also need to be provided proper annotations for those. and also need to specify every element that is presented in the graphs.

6.        The novelty of the proposed method is confusing and the contribution of this method to the system is to be specified clearly with required particulars.  

7.        The terminology which Is used in the manuscript is to be clear and should not make the confusions for the readers. So be clear with the usage of the terminologies used for the manuscript.

8.        The data provided in the tables are clumsy so carefully categorize and provide clearly to differentiate the data from each category.

9.        The paper is so clumsy and it makes readers get confused. So kindly provide clarity in the manuscript. Because some unwanted subsections are there and the connectivity between subsections is missing. Ensure that the linking of the sections is mandatory for the manuscript for better understanding. 

10.    The authors needed to concentrate on the abbreviations used. Because few words were not even explained in the manuscript. Needs to be used proper abbreviations.

11.    In section 4, The data provided is not accurate. Need more accurate data and the authors need to provide more concentration on that section and also need to avoid the typo errors in that section as well as the entire manuscript.

12.    The conclusion part is very lengthy and it needs to be summarized. And be specific with the data provided for the conclusions part.

13.    The references are to be revised once and the citations provided for the references are to be in the proper manner in the entire manuscript. Some citations were not in the proper manner so careful revision is required.    

14.    The overall revision is required for the manuscript for arranging the data in proper sections and subsections. It helps the readers to study in a clear and understanding way.

15.    Why do authors use TTC as the evaluation metrics? What about the TTE and TTR? What difference does it make if we go with TTE and TTR for evaluation purposes?

16.    What about other methods for evaluating multiple scenarios of AVs? How your method is different from those and what makes your method superior to them?

17.    The authors provided the differences and advantages of multiple scenarios evaluation over a single scenario. That’s good but the comparison with other multiple-scenario evaluation methods is required for justifying the proposed method is better. Authors are required to concentrate on this aspect.  

Minor editing of English language required

Author Response

The authors deeply appreciate the reviewers’ valuable comments and suggestions, which greatly helped to improve the quality of this paper.

In the attached file, we present a point-by-point list of how we addressed each of the reviewers’ comments and suggestions.

Again, we would like to thank the anonymous reviewers and the editor for their constructive and insightful comments. We have incorporated all the comments into this revised manuscript, which has resulted in a more concise and greatly improved paper.

Sincerely yours,

Soomok Lee

Reviewer 2 Report

Thank you very much for submitting your manuscript to Sensors. Automated driving systems (ADSs) recently is one of the most attractive research topic of a lots researchers and manufacturers. With the aim at aiding in continuously evaluating the ADS performance over entire road segments, this paper proposed a methodology that combines and transforms single scenarios into multiple scenarios. Although this study is also towards the new trend nowadays; however, from the viewpoint of theoretical development, the authors need more significant methodologies and comprehensive comparison of different methods. Besides, there are some comments need to be clarified as describe in the following.

Comments:

1.    In the Literature Review, the authors presented the related theory, prior research, and lesson learned to point out the advantages and disadvantages of the previous study and then insist that the proposed method outperforms. However, from this section, the reviewer is not convinced that this work provided added value to the state of the art. The authors should synthesize the contributions of the proposed study into a list for convenient review.

2.  As mentioned in the manuscript, a single scenario limitation in comprehensively evaluating the performance of complex ADS. Then, the multiple scenarios which is combined and transformed form single scenarios will aid in continuously evaluating the ADS performance. Nevertheless, how to evaluate this advantage?

3.    In order to valuate the multiple scenarios, this research presented Time-to-collision (TTC) simulation to make the comparison. TTC refers to the time required for two vehicles to collide if they continue driving at their current velocity. Thus this feature is independent of applying the single scenario or the multiple scenarios. It is mainly governed by the vehicle’s control algorithm. Therefore, the authors should clarify the effect and applicability of the proposed research as well.

4.     There is a room for improving the English of the manuscript as there are some typos, grammatical mistakes, writing style for a scientific research paper. Would be good for the authors to review the paper thoroughly and correct the English.

There is a room for improving the English of the manuscript as there are some typos, grammatical mistakes, writing style for a scientific research paper. Would be good for the authors to review the paper thoroughly and correct the English.

Author Response

Dear Reviewer,

The authors deeply appreciate the reviewers’ valuable comments and suggestions, which greatly helped to improve the quality of this paper.

In the attached file, we present a point-by-point list of how we addressed each of the reviewers’ comments and suggestions.

Again, we would like to thank the anonymous reviewers and the editor for their constructive and insightful comments. We have incorporated all the comments into this revised manuscript, which has resulted in a more concise and greatly improved paper.

Sincerely yours,

Soomok Lee

Round 2

Reviewer 1 Report

My comments were addressed properly 

Minor changes needed 

Author Response

(The authors gave the same response as above.)

Reviewer 2 Report

Thank you very much for accepting my comments and revising the manuscript. However, there are still some questions that need to be clarified more thoroughly.

Comments:

1. The authors explained the performance evaluation of multiple scenarios via TTC and give some references. However, I can not find Schwarz (2014). Please carefully check the manuscript and the revised answer.

2.  In section 4.4, the authors designed a simulation to compare single scenario and multiple scenarios. The setting is presented in 4.4.1 which includes scenario 1 and scenario 2. Therefore, the authors evaluate the performance of the proposed method based on TTC of ego vehicle in the toll gate section. Why do you choose this section? What is the control approach of the ego and the surrounding vehicles? Furthermore, Fig.7 is TTC results. The period of time is considered under 3 secs. It is too short to accurately represent the outperform of the proposed method. It would be more convincing if the authors can present the TTC result in the whole scenario or at least at each section, i.e. ramp section, main section, and toll gate section.

Author Response

(The authors gave the same response as above.)
